# Prevalence of blindness and its determinants in Bangladeshi adult population: results from a national cross-sectional survey

Shawkat Ara Shakoor [1], Mustafizur Rahman [2], A H M Enayet Hossain [3], Mohammad Moniruzzaman [4], Mahfuzur Rahman Bhuiyan [5], Ferdous Hakim [6], M Mostafa Zaman [6]

¹Community Ophtalmology, National Institute of Ophthalmology, Dhaka, Bangladesh
²Ophthalmology, Dhaka Medical College, Dhaka, Bangladesh
³Paediatric Ophthalmology, National Institute of Opthalmology, Dhaka, Bangladesh
⁴Public Health, Shiga University of Medical Science, Otsu, Japan
⁵Epidemiology and Research, National Heart Foundation Hospital and Research Institute, Dhaka, Bangladesh
⁶Research and Publication, World Health Organization, Dhaka, Bangladesh

**Correspondence to**
Dr M Mostafa Zaman;
zamanm@who.int

## ABSTRACT

**Objective** The objective of this study was to determine the prevalence of blindness and its determinants in Bangladeshi adult population.

**Study design** A cross-sectional population-based survey conducted at household level with national representation. Samples were drawn from the 2011 national census frame using a multistage stratified cluster sampling method.

**Setting and participants** The survey was done in urban and rural areas in 2013 using a probability proportionate to size sampling approach to locate participants from 72 primary sampling units. One man or one woman aged ≥40 years was randomly selected from their households to recruit 7200. In addition to sociodemographic data, information on medication for hypertension and diabetes was obtained. Blood pressure and capillary blood glucose were measured. Eyelids, cornea, lens, and retina were examined in addition to visual acuity and refraction testing.

**Primary outcome measures** The following definition was used to categorise subjects having (1) blindness: visual acuity <3/60, (2) low vision: ≥3/60 to <6/60 and (3) normal vision: ≥6/12 after best correction.

**Results** We could recruit 6391 (88.8%) people among whom 2955 (46.2%) were men. Among them, 1922 (30.1%) were from urban and 4469 (69.9%) were from rural areas. The mean age was 54.3 (SD 11.2) years. The age-standardised prevalence, after best correction, of blindness and low vision was 1.0% (95% CI 0.5% to 1.4%) and 12.1% (95% CI 10.5% to 13.8%), respectively. Multivariable logistic regression indicated that cataract, age-related macular degeneration and diabetic retinopathy were significantly associated with low vision and blindness after adjustment for age and sex. Population attributable risk of cataract for low vision and blindness was 79.6%.

**Conclusions** Low vision and blindness are common problems in those aged 40 years or older. Extensive screening and eye care services are necessary for wider coverage engaging all tiers of the healthcare system especially focusing on cataract.

## BACKGROUND

The impact of visual loss on an individual's personal, economic and social life is profound. When the burden of blindness

## Strengths and limitations of this study

► This nationally representative population-based survey indicates that more than 1 in 10 Bangleashi adults aged ≥40 years have low vision or blindness, with cataract being the single most attributing factor.
► The study followed rigorous survey methods, including a multistage, geographically clustered and probability proportional to size sampling approach to recruit particiapnts randomly.
► The absence of colour photos of fundus examinations might have led to biased estimate of age-related macular degeneration and diabetic retinopathy.

in communities is high, the consequences become a significant public health issue.[1] According to the WHO, 285 million people globally lived with visual impairment in 2010. Of them, 246 million had low vision; 39 million were blind; and two-thirds of this population were aged over 50 years.[2] Because of the rapid population ageing, low vision and blindness have become a global public health threat, particularly in low-income countries.

Nearly 90% of the world's visually impaired people live in low-income countries. The Southeast Asia Region, including Bangladesh, is estimated to inhabit 90.5 million visually impaired and 12 million blind adults in 2010.[3] Globally, the top four causes of visual impairment are uncorrected refractive errors, cataract, age-related macular degeneration (AMD) and glaucoma. Therefore, 80% of all visual impairments are avoidable.[3]

In Bangladesh, a previous national survey—done in 2000—reported an age-standardised prevalence of blindness and low vision of 1.53% and 0.56%, respectively, among adults aged 30 years or older.[4 5] Since then, Bangladesh has passed through a remarkable demographic transition. Recent data on blindness

and low vision in Bangladesh are unknown. Bangladesh has been implementing its National Eye Care for preventing avoidable blindness and low vision, but mostly through tertiary level hospitals.[6] A recent estimate, therefore, was required to inform the eye care plan and other relevant programmes. We conducted this national survey to determine the prevalence of blindness and impaired vision, and related factors in Bangladeshi adults.

## METHODS

### Study design, population and setting

We conducted a nationwide population-based cross-sectional survey among Bangladeshi adults (men and women) aged 40 years or older in September—December 2013. We calculated our sample size based on a prevalence of blindness (1.53%), with a margin of error (0.00765) and a design effect of 1.5 (1483). Then we adjusted for four groups (men, women, urban and rural) and a response rate of 82.5% (7193), leaving the final sample size to 7200. The details of the sampling procedure have been described previously.[7] Briefly, we adopted a multistage, geographically clustered, probability-based sampling approach to obtain a nationally representative sample. We invited a total of 7200 randomly selected adults from 72 (urban, 25; rural, 47) primary sampling units (used in the 2011 national census) to participate from all seven divisions of Bangladesh. In each selected primary sampling unit, we identified 100 consecutive households with a random start. Then we randomly selected one participant from a list of eligible household members using the Kish table.[8] The flowchart of subject selection is given in figure 1.

### Patient and public involvement

Patients and the public were not involved in this study.

### Training of the survey team

The survey team was composed of experienced enumerators, ophthalmic nurses, medical technologists and ophthalmologists. They were trained in the National Institute of Ophthalmology by the investigators. On completion of their training, a dry-run was given in two nearby rural and urban areas. They were trained (as a team) using a using a study manual before launching the survey to reduce interobserver variations and to improve diagnostic accuracy. Their findings were randomly checked by the investigators at least once in each primary sampling unit.

### Data collection

As depicted in figure 1, trained enumerators collected demographic, socioeconomic and medical history data using an interviewer-administered standardised questionnaire at the household level. Thereafter, they invited participants to have a physical and ophthalmic examination in a nearby health centre (or makeshift examination centre established conveniently by the research team).

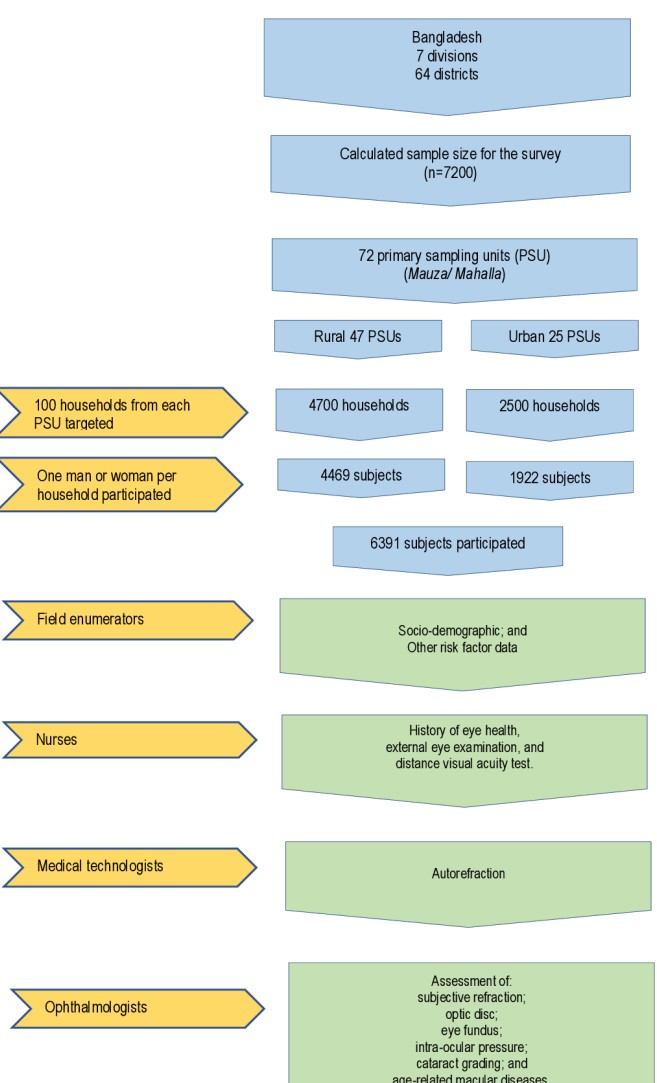

**Figure 1** Flowchart for subject selection of the cross-sectional national survey done in urban and rural areas of all seven divisions in Bangladesh (n=6391).

Nurses measured participants' height, weight, seated blood pressure and (random) capillary blood glucose using a glucometer (Accu-chek Advantage; Roche Diagnostics Division, Switzerland). We used a modified WHO/PBL questionnaire V.3[9] as our instruments.[10]

### Vision and ophthalmic examinations

We used WHO International Classification of Diseases 10 categories of visual impairment for the study.[11 12] Blindness was defined as corrected visual acuity of less than 3/60 in the better eye. Low vision was defined as corrected visual acuity of less than 6/60 but equal to or more than 3/60 in the better eye. People having visual acuity of 6/12 or more were considered to have normal vision.

Eyelids, cornea, lens (including its absence or displacement) and retina were examined. AMD was defined as the presence of any one of the following: soft drusen or reticular drusen, hyperpigmentation or hypopigmentation of the retinal pigment epithelium. Diabetic retinopathy included non-proliferative, proliferative and

maculopathy subtypes. These were not mutually exclusive as the latter two types, for example, may coexist.

Hypertension was defined as blood pressure ≥140/90 mm Hg or use of antihypertensive medicines, and diabetes was defined as casual capillary blood glucose of ≥11.1 mmol/L or use of antidiabetic medicines. Distance visual acuity was measured on unaided participants with Snellen 'E' chart and a hand-held tally counter, if necessary, at 3 m by ophthalmic nurses. Depending on acuity, finger count, hand movement and light projections were used. Medical technologists have done autorefraction. Thereafter, subjective refractions were done by the ophthalmologists. Based on presenting visual acuity, participants were assigned either a red card (acuity worse than 6/12 in either eye) or a green card (equal or better than 6/12 in both eyes tested separately).

Intraocular pressure was measured using Schiotz tonometer after application of tetracaine hydrochloride (1%). A relative afferent pupil defect in those patients with a best-corrected visual acuity of <6/12 in either eye was tested. The ophthalmologist assessed the fundus, including optic disc, cup/disc ratio, macula in both eyes using a direct ophthalmoscope through an undilated pupil. All participants with a best-corrected visual acuity of less than 6/12 were subsequently dilated, and the fundus re-checked with an indirect ophthalmoscope. A compound solution of tropicamide (1%) was used to obtain a pupil diameter of at least 6 mm. Those deemed at risk of angle-closure (following an oblique flashlight test) were not dilated. Those with the vertical cup:disc ratio of ≥0.70 in either eye in the presence of intraocular pressure of ≥97.5 percentile were identified as having glaucoma.[13]

### Data analysis

Data were analysed using Microsoft Excel (Microsoft Office 365) and Epi Info V.7.1.2.5 after necessary cleaning and logical checks. Age was categorised into two groups: 40–54 years and ≥55 years. We estimated the prevalence of mild, moderate and severe impaired vision and blindness (as described earlier) with 95% CIs. We presented the main results stratified by four reporting domains: residence location (urban–rural) and sex (men–women). Age adjustment of prevalence estimates was done based on WHO World Population 2000–2020.[14]

Factors associated with impaired vision and blindness were checked with 2×2 cross-tabulation. Unadjusted ORs were obtained by univariate logistic regression analysis. Finally, risk factors independent of age and sex were identified using multiple logistic regression. Age and sex were entered into all the models. Thus, adjusted ORs and their 95% CIs were obtained to check the strengths of the association. At the same time, p values less than 0.05 were also noted for convenience.

### RESULTS

We recruited 6391 persons out of the targeted 7200, resulting in a response rate of 88.8%. Among the respondents, 3436 (53.8%) were women (table 1). Men and women were similar in terms of age categories and average (54.3 years with an SD of 11.2 years). Half (50.9%) of them never attended formal school, and one-fifth (21.9%) had above primary education. Women mainly were homemakers (79.2%), but almost half (48.6%) of men were manual workers. More than 6 in 10 (63.6%) were tobacco (smoking or smokeless) users. However, there was hardly anyone with an alcohol drinking habit (1.2%). One-fifth (20.5%) were overweight (body mass index ≥25.0 kg/m$^2$); 25.4% had hypertension (blood pressure ≥14/90 mm Hg or medication); and 7.8% had diabetes mellitus (random blood glucose ≥11.1 mmol/L or on medication for diabetes).

### Low vision and blindness

The prevalence of corrected visual acuity by age, sex and residence is given in table 2. Overall, the age-adjusted prevalence of low vision and blindness was 12.1% and 1.0%, respectively. Blindness was higher in those aged 55 years or older (1.8%) compared with the younger people (0.2%) (<55 years old). Further splitting of age showed an increasing trend of blindness prevalence across age groups (figure 2). No differences were observed between sexes and residential areas, as indicated by the overlapping 95% CIs (table 2).

### Factors associated with low vision and blindness

In our sample, 22.9% (95% CI 18.7% to 24.6%) had had cataract of some form; 1.7% (95% CI 1.2% to 2.3%) had diabetic retinopathy; 0.8% (95% CI 0.5% to 1.2%) had glaucoma; 0.8% (95% CI 0.5% to 1.1%) had corneal diseases; 0.5 (95% CI 0.3% to 0.7%) had AMD; and 0.4 (95% CI 0.2% to 0.6%) had eyelid disorders (figure 3). Altogether, 84.3% of patients with low vision and blindness had cataract (table 3). Univariate logistic regression indicated a significant relationship of low vision and blindness with age, male sex, cataract, diabetic retinopathy, glaucoma and AMD. However, multiple logistic regression after adjusting for age and sex showed a significant association, in order of strength, of cataract (OR 17.0, 95% CI 13.7 to 21.2), AMD (OR 5.2, 95% CI 2.1 to 12.7) and diabetic retinopathy (OR 2.2, 95% CI 1.4 to 3.5) (table 3). Cataract's attribution to blindness was the largest among all. Population attributable risk of cataract for blindness was 79.6%.

### DISCUSSION

We report here findings of the second national-level survey, done after 13 years of the first national survey[5] done in 2000, that age-adjusted prevalence of blindness in Bangladeshi adults is 1.0% after best possible correction of vision. This estimate is lower than that reported by the first national survey (1.53%).[5] However, it is important to note that the first survey was done among those aged 30 years or older. Younger people are expected to have a lower burden of blindness. The ageing of the Bangladeshi

**Table 1** Sociodemographic characteristics and relevant risk factors of the respondents, n (%)

| Variables | Both (n=6391) | Men (n=2955) | Women (n=3436) |
|---|---|---|---|
| Age group (years)* | | | |
| <55 | 3684 (57.6) | 1642 (55.6) | 2042 (59.4) |
| ≥55 | 2707 (42.4) | 1313 (44.4) | 1394 (40.6) |
| Residence | | | |
| Urban | 1922 (30.1) | 841 (28.5) | 1081 (31.5) |
| Rural | 4469 (69.9) | 2114 (71.5) | 2355 (68.5) |
| Education | | | |
| No formal schooling | 3238 (50.9) | 1147 (38.9) | 2091 (61.1) |
| Any primary (classes 1–5) | 1733 (27.2) | 862 (29.3) | 871 (25.5) |
| Above primary (classes ≥6) | 1397 (21.9) | 937 (31.8) | 460 (13.4) |
| Occupation | | | |
| Professional employee† | 1015 (15.9) | 886 (30.1) | 129 (3.8) |
| Industrial worker/day labourer | 1587 (24.9) | 1430 (48.6) | 157 (4.6) |
| Homemaker | 2716 (42.6) | 0 (0.0) | 2716 (79.2) |
| Unemployed/retired | 901 (14.1) | 503 (17.1) | 398 (11.6) |
| Others‡ | 153 (2.4) | 124 (4.2) | 29 (0.8) |
| Tobacco use (smoking or smokeless) | 4066 (63.6) | 2122 (71.8) | 1944 (56.6) |
| Alcohol use, last 30 days | 77 (1.2) | 69 (2.3) | 8 (0.2) |
| Overweight/obesity§ | 1300 (20.5) | 455 (15.5) | 845 (24.7) |
| Diabetes mellitus¶ | 498 (7.8) | 230 (7.8) | 268 (7.8) |
| Hypertension** | 1623 (25.4) | 689 (23.3) | 934 (27.2) |

Missing data for education, 23; occupation, 19; current tobacco use, 15; alcohol use in last 30 days, 21; body mass index, 32; diabetes mellitus, 8.

*Cut-off based on mean age (54.3 years).

†Professional employment: government and private company employees and businessmen.

‡Others: shopkeeper, weaver, driver, beggar, cook, carpenter and tailor.

§Body mass index ≥25 kg/m$^2$; one pregnant woman was excluded.

¶Diabetes mellitus: random capillary blood glucose ≥11.1 mmol/L and/ or known history of diabetes; one pregnant woman was excluded.

**Hypertension: blood pressure ≥140/90 mm Hg or on medication for hypertension.

**Table 2** Prevalence (%) of corrected visual acuities, per cent (95% CI)

| Characteristics | Number (n=6391) | Normal (≥6/12) (n=5628) | Low vision (≥3/60 to <6/60) (n=707) | Blind (<3/60) (n=56) |
|---|---|---|---|---|
| Age group (years) | | | | |
| <55 | 3684 | 98.1 (97.6 to 98.6) | 1.7 (1.2 to 2,2) | 0.2 (0.01 to 0.4) |
| ≥55 | 2707 | 74.4 (71.4 to 77.4) | 23.8 (20.9 to 26.7) | 1.8 (1.1 to 2.5) |
| Sex | | | | |
| Men | 2955 | 87.2 (85.1 to 89.3) | 12.0 (10.0 to 14.1) | 0.7 (0.4 to 1.1) |
| Women | 3436 | 88.8 (87.4 to 90.2) | 10.2 (8.9 to 11.6) | 1.0 (0.6 to 1.4) |
| Residence | | | | |
| Urban | 1922 | 87.7 (85.2 to 90.3) | 11.8 (9.2 to 14.3) | 0.5 (0.2 to 0.9) |
| Rural | 4469 | 88.2 (86.3 to 90.1) | 10.8 (8.9 to 12.6) | 1.0 (0.6 to 1.4) |
| Overall | 6391 | 88.1 (86.5 to 89.5) | 11.1 (9.6 to 12.6) | 0.9 (0.6 to 1.2) |
| Overall (age adjusted) * | | 86.9 (85.2 to 88.6) | 12.1 (10.5 to 13.8) | 1.0 (0.5 to 1.4) |

*Adjusted for WHO World Population 2000–2020.[14]

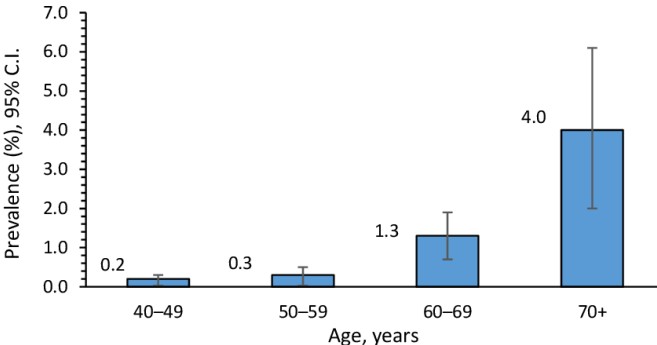

**Figure 2** Prevalence of blindness according to age groups among the respondents of the cross-sectional national survey on visual impairments in Bangladesh (n=6391).

population is well known because of the demographic transition.[15] Moreover, the national eye care programme intervention might have contributed to this decline in blindness prevalence. The national eye care plan[4] emphasised activities to reduce blindness focusing cataract surgery that is low-cost, organising outreach camps for screening, awareness creation and manpower training. The plan facilitated establishment of treatment centres at district level and eyesight testing through partnership of government and non-governmental organisations.

### Prevalence

The prevalence of blindness in Singapore (0.4%),[16] Taiwan (0.6%),[17] Malaysia (0.3%),[18] China (0.3%)[19] and USA (0.5%)[20] is similar to the prevalence we report here (1.0%). There was a wide variation of prevalence of blindness in Asian countries like Pakistan is 2.7%,[21] Mongolia (1.5%),[22] rural Indonesia (2.2%),[23] India (5.3%),[24] Nepal (1.9%),[25] Nigeria (4.2%)[26] and Iran (1.1%).[27] These variations, however, may be due to differences in the definition of blindness used in the surveys, age composition of the sample and survey design. Increasing trend of blindness and visual impairment with age in our sample is somewhat similar to surveys done in India[24] and Iran.[27] Unlike our survey, Pakistan reported a higher prevalence in the rural population and in women.[21] Malaysia also reported a higher prevalence in women compared with men.[18] Nonetheless, no sex difference was found in the Taiwanese population.

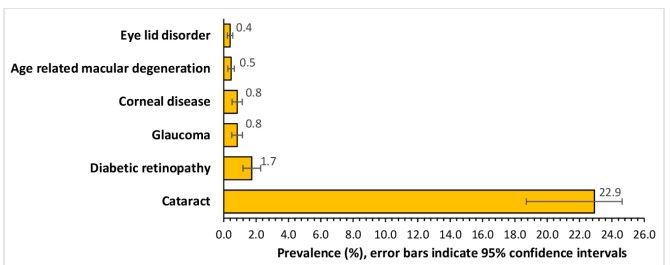

**Figure 3** Prevalence of various eye conditions among the respondents of the cross-sectional national survey on visual impairments in Bangladesh (n=6391).

Apparently, we observed a higher prevalence of low vision (12.1%) compared with studies in India (9.3%),[24] Pakistan (3.3)[21] and Iran (4.0%),[27] but it was somewhat similar to that reported from South American countries (5.9%–12.5%).[28] These differences should be cautiously interpreted because variation in age composition of the respondents, as well as some other factors, is an important determinant of low vision.

### Associated factors/causes

We identified cataract, AMD and diabetic retinopathy as the major causes of blindness in our population. Cataract's attribution to blindness was the largest among all. Cataract is the leading cause of blindness worldwide and is responsible for 94 million blindness.[3] This is true for Asian countries,[4 16–19 21–25 27] including Bangladesh.[5] The leading causes of visual impairment in the Taiwanese population are cataract, amblyopia due to uncorrected refractive errors, vitreoretinal diseases, corneal blindness and diabetic retinopathy.[17] In Singapore across all ethnic groups, cataract was the leading cause of bilateral blindness. Other major causes of blindness included diabetic retinopathy, AMD, glaucoma, corneal opacity and myopic maculopathy.[16] In Western countries, AMD is the main cause of blindness, especially after the age of 50 years.[29] Diabetic retinopathy, as we observed, was an important factor for blindness in Taiwan[30] and in many states of India.[31–33] However, all the comparison we show here are very much dependent on age and sex of the participating subjects and therefore should be interpreted with caution.

Cataract's attribution to blindness in our sample (79.6%) is a little higher than that reported in an India population (62.1%).[34] Therefore, addressing cataract will be bring most benefit to prevent blindness. In addition to promotion of healthy ageing, a few other factors such as ultraviolet ray exposures, diabetes, hypertension, use of certain drugs and smoking can be considered.[35 36] Accessibility to socioeconomically deprived people especially in remote areas should be enhanced. Blindness prevention programmes' success will largely depend on the health system's capacity building to deliver low-cost cataract surgeries. Supplementation from outreach screening will be valuable.

This study has its inherent strength that sample has a national representation, which was drawn from the primary sampling units used by the national statistical authority. It was done by employing a multidisciplinary team that included professional enumerators, ophthalmic nurses, medical technologists and ophthalmologists. The study, on the other hand, has some limitations too. We could not have colour photos of fundus examinations for subsequent validation of findings. Therefore, some degree of underestimation of AMD and diabetic retinopathy diagnoses cannot be overruled.

### CONCLUSIONS

This study provides essential information on blindness burden and its prevention in Bangladesh. The

**Table 3** ORs of risk factors for impaired vision and blindness after correction in Bangladeshi adults (n=6391)

| Factors | | Vision categories | | OR (95% CI) | |
|---|---|---|---|---|---|
| | | Low vision and blind (<6/12) (n=763) | Normal vision (≥6/12) (n=5628) | Unadjusted | Adjusted for age and sex |
| Age (years) | ≥55 | 693 (90.8) | 2014 (35.8) | 17.8 (13.8 to 22.9)* | – |
| (≥55=1, <55=0) | <55 | 70 (9.2) | 3614 (64.2) | 1.0 | – |
| Sex | Men | 378 (49.5) | 2577 (45.8) | 1.2 (1.0 to 1.4)* | – |
| (man=1, woman=0) | Women | 385 (50.5) | 3051 (54.2) | 1.0 | – |
| Diabetes mellitus† | Yes | 64 (8.4) | 435 (7.7) | 1.1 (0.8 to 1.4) | 1.0 (0.7 to 1.3) |
| (yes=1, no=0) | No | 698 (91.6) | 5186 (92.3) | 1.0 | 1.0 |
| Hypertension | Yes | 192 (25.2) | 1431 (25.4) | 1.0 (0.8 to 1.2) | 0.8 (0.6 to 0.9) |
| (yes=1, no=0) | No | 571 (74.8) | 4197 (74.6) | 1.0 | 1.0 |
| Cataract | Yes | 643 (84.3) | 822 (14.6) | 31.3 (25.4 to 38.6)* | 17.0 (13.7 to 21.2)* |
| (yes=1, no=0) | No | 120 (15.7) | 4806 (85.4) | 1.0 | 1.0 |
| Diabetic retinopathy | Yes | 31 (4.1) | 80 (1.4) | 2.9 (1.9 to 4.5)* | 2.2 (1.4 to 3.5)* |
| (yes=1, no=0) | No | 732 (95.9) | 5548 (98.6) | 1.0 | 1.0 |
| Glaucoma | Yes | 13 (1.7) | 40 (0.7) | 2.4 (1.3 to 4.5)* | 1.4 (0.7 to 2.7) |
| (yes=1, no=0) | No | 750 (98.3) | 5588 (99.3) | 1.0 | 1.0 |
| AMD‡ | Yes | 12 (1.6) | 17 (0.3) | 5.3 (2.5 to 11.1)* | 5.2 (2.1 to 12.7)* |
| (yes=1, no=0) | No | 751 (98.4) | 5611 (99.7) | 1.0 | 1.0 |
| Corneal disease | Yes | 6 (0.8) | 47 (0.8) | 0.9 (0.4 to 2.2) | 0.9 (0.4 to 2.4) |
| (yes=1, no=0) | No | 757 (99.2) | 5581 (99.2) | 1.0 | 1.0 |
| Ocular trauma | Yes | 3 (0.4) | 7 (0.1) | 3.2 (0.8 to 12.3) | 3.4 (0.7 to 16.6) |
| (yes=1, no=0) | No | 760 (99.6) | 5621 (99.9) | 1.0 | 1.0 |
| Eyelid disorder | Yes | 4 (0.5) | 21 (0.4) | 1.4 (0.5 to 4.1) | 0.6 (0.2 to 1.9) |
| (yes=1, no=0) | No | 759 (99.5) | 5607 (99.6) | 1.0 | 1.0 |

*P<0.01
† 8 missing values.
‡ AMD: age related macular degeneration.

age-adjusted prevalence of blindness in Bangladesh is approximately 1% in adults aged 40 years or older. Cataract, AMD, glaucoma and diabetic retinopathy are the major factors for blindness. The attribution of cataract outweighs all others, being responsible for 80% of the preventable causes. Given that national eye care is primarily based in tertiary care hospitals, we recommend strengthening primary and secondary care systems to reach out to most people who need the services. The creation of public awareness for seeking services could broaden the coverage of national eye care.

**Acknowledgements** Our gratitude goes to Professor Deen Mohd, Noorul Huq and Professor Jalal Ahmed for their guidance, and to Dr Mohd Abdullah Al Mamun for his support. We acknowledge the contribution provided by Drs Masum Habib, Md Abdul Quader, Zahid Ahsan Mennon, Iftekhar Md Munir and Md Shahabuddin for leading the field team for data collection. We thank Ms Khaleda Akter for her assistance in preparing the reference list, formatting the document and obtaining the necessary approval for publication.

**Collaborators** Our gratitude goes to Professor Deen Mohd. Noorul Huq and Professor Jalal Ahmed for their guidance, and to Dr Mohd. Abdullah Al Mamun for his support.

**Contributors** The study was conceptualised by SAS, MR, AHMEH and MMZ. The literature review was accomplished by MM and MRB. Study design and sampling were prepared by MMZ, MM and MRB. The questionnaire was developed and tested by SAS and MRB. The training manual was drafted, and enumerators were trained by MM, MRB and MMZ. All investigators took part in the data collection, supervision and quality assurance measures. Data cleaning and analysis were done by FH and MM with the guidance of MMZ. MMZ critically interpreted the results. SAS conceptualised and prepared the first draft of the manuscript, which was critically reviewed, revised and finalised by MMZ, MM, MRB and FH. AHMEH is the guarantor of data. MMZ guided the whole study.

**Funding** The WHO Country Office for Bangladesh provided financial assistance for this study (WHO Reference: 2013/355662-0, Purchase Order: 200843353, Reg. File: BAN-2013-B7-TSA-0001). However, no fund was used in preparing this manuscript.

**Competing interests** None declared.

**Patient and public involvement** Patients and/or the public were not involved in the design, conduct, reporting or dissemination plans of this research.

**Patient consent for publication** Not applicable.

**Ethics approval** This study involves human participants, and ethical approval was obtained from the institutional review board of the National Institute of

Ophthalmology, Dhaka, Bangladesh (memo number NIO/670, 4 April 2013). Participants gave informed consent to participate in the study through signature and, if not possible, through thumbprint.

**Provenance and peer review** Not commissioned; externally peer reviewed.

**Data availability statement** Data are available upon reasonable request. Please contact Professor M Mostafa Zaman, emails: zamanm@who.int and mmostafazaman@gmail.com.

**ORCID iDs**
Shawkat Ara Shakoor http://orcid.org/0000-0001-6299-7529
Mustafizur Rahman http://orcid.org//0000-0002-7199-1070
A H M Enayet Hossain http://orcid.org/0000-0003-2715-5574
Mohammad Moniruzzaman http://orcid.org/0000-0003-2144-7111
Mahfuzur Rahman Bhuiyan http://orcid.org/0000-0001-6962-7264
Ferdous Hakim http://orcid.org/0000-0003-2376-3978
M Mostafa Zaman http://orcid.org/0000-0002-1736-1342

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
