## [Reviewer comments · BMJ Open]

ARTICLE DETAILS

TITLE (PROVISIONAL)	Prevalence of blindness and its determinants in Bangladeshi adult population: Results from a national cross-sectional survey
AUTHORS	Ara Shakoor, Shawkat; Rahman, Mustafizur; Hossain, A.H.M. Enayet; Moniruzzaman, Mohammad; Bhuiyan, Mahfuzur; Hakim, Ferdous; Zaman, MM

VERSION 1 – REVIEW

REVIEWER	You, Qisheng Oregon Health & Science University
REVIEW RETURNED	24-Jul-2021

GENERAL COMMENTS	This large-scale population-based study addressed the prevalence and associations of visual impairment and blindness in Bangladesh. There is a paucity of ophthalmic epidemiological data from Bangladesh. Therefore, it may be worth publishing. However, there are obvious limitations in study methods, which currently are not fully addressed in the manuscript. (1) Is refraction performed in the study? If not, the vision won't be best corrected. This needs to be discussed. If yes, how was the refraction performed? Please clarify. (2) It is also unclear how the authors defined presenting visual acuity. Please clarify. (3) The methods to detect glaucoma and AMD, DR is not standardized. The current methods (examined by clinicians using direct or indirect biomicroscopy) may lead to the under-detection of these diseases, hence under-estimate the prevalences. These methods are also not verifiable. (4) The limitations in disease diagnosis methods caused marked difficulty in interpreting the results.
--

REVIEWER	Xiao, Baixiang Zhongshan Ophthalmic Centre, Sun Yat-sen University, State Key Laboratory of Ophthalmology
REVIEW RETURNED	26-Jul-2021

GENERAL COMMENTS	The study reported valuable findings from a national blindness and low vision survey from the country and it is very important to inform future work across the country. There are some uncertainty as follows: Abstract: P4, line 31, definition of low vision should be modified. Line 34 – 37, percentage should be given for each of the numbers. Line 48 – 51 good conclusion and recommendations.
--

	The authors say that is a national representative study, how was this represented? Are all the people in Bangladesh in the population pool listed for the randomly sampling? The article did not articulate. Background: P5, line 8-11 Please give the time estimated. P6, line 33-34, what were the specific national strategies for preventing avoidable blindness? P6 – 7: sampling: what the census used? Are all people from the country over 40 years old in the sample framework? How many clusters/size of clusters were selected for the 7,200 people. P9, line 36-40 and P13, table 1, p-value should be given for each of the characteristics to tell the significance. P9, line 47-53, these percentage should have confidence interval P10, discussion: as the age specific prevalence of BL/LV were not presented, so the comparison of prevalence due to ageing is not clear. P10, line 27-28, your prevalence was 12.1%, how could this be the same as the countries listed? With less than 1%? P10, line 35 -41 , Cataract is the major cause of blindness, the reasons of low prevalence in other countries that the authors given could not convince readers. In general, the discussion is not robust.
--	--

VERSION 1 – AUTHOR RESPONSE

Reviewers' comments on Manuscript ID bmjopen-2021-052247

REVIEWER: 1

This large-scale population-based study addressed the prevalence and associations of visual impairment and blindness in Bangladesh. There is a paucity of ophthalmic epidemiological data from Bangladesh. Therefore, it may be worth publishing. However, there are obvious limitations in study methods, which currently are not fully addressed in the manuscript.

>>Thank you very much for your encouraging comments.

1. Is refraction performed in the study? If not, the vision won't be best corrected. This needs to be discussed. If yes, how was the refraction performed? Please clarify.

>> Yes. Auto refraction was done, and then subjective refraction was performed. This has been shown in Figure 1, and relevant texts added (page 6, line 23)

2. It is also unclear how the authors defined presenting visual acuity. Please clarify.

>> Distance visual acuity was measured on unaided participants with Snellen 'E' chart and a hand-held tally counter, if necessary, at three meters by ophthalmic nurses. Depending on acuity, finger count, hand movement and light projections were used. Medical technologists have done autorefraction. Thereafter, subjective refractions were done by the ophthalmologists. (page 6, line 19-23)

3. The methods to detect glaucoma and AMD, DR is not standardized. The current methods (examined by clinicians using direct or indirect biomicroscopy) may lead to the under-detection of these diseases, hence under-estimate the prevalence. These methods are also not verifiable.

>> We have added a sub-section to address this issue as below: (page 5, line 10-17)

Training of the survey team

The survey team was comprised of experienced enumerators, ophthalmic nurses, and medical technologists and ophthalmologists. They were trained in the National Institute of Ophthalmology by the investigators. Upon completion of their training, a dry-run was given in two nearby rural and urban areas. They were trained (as a team) using a study manual before launching the survey to reduce inter-observer variations and improve diagnostic accuracy done by the ophthalmologists. Their findings were randomly checked by the investigators at least once in a primary sampling unit.

In a large-scale survey in low resource settings such approach is being used. However, we acknowledge the possibility of an under-estimation in the Limitation sub-section (page 10, line 10).

4. The limitations in disease diagnosis methods caused marked difficulty in interpreting the results.

>>We acknowledged the limitation of diagnosis AMD and DR in addition to the non-availability of color photograph of the fundus. Possibility of an under-estimation has been added (page 10, line 10). Otherwise, our diagnoses were accurate and reliable.

REVIEWER: 2

The study reported valuable findings from a national blindness and low vision survey from the country and it is very important to inform future work across the country.

>>Thank you very much for your encouraging comments.

There are some uncertainties as follows:

1. **Abstract:** P4, line 31, definition of low vision should be modified.

>>The definition is same, but we change our presentation: $\geq 3/60$ - $< 6/60$

2. Line 34 – 37, percentage should be given for each of the numbers.

>>Percentages added

3. Line 48 – 51 good conclusion and recommendations.

>>Thank you

4. The authors say that is a national representative study, how was this represented? Are all the people in Bangladesh in the population pool listed for the randomly sampling? The article did not articulate.

>>This is nationally representative survey using a multistage cluster sampling as per the principle of PPS (page 4, line 33 continued to page 5 line 1-5, and the flowchart given in Figure 1). The study used 72 primary sampling units (clusters) of the national census. (page 5, line 1)

5. P5, **Background**: line 8-11 Please give the time estimated.

>>Time added

6. P6, line 33-34, what were the specific national strategies for preventing avoidable blindness?

>> The following texts have been added to the Discussion. (page 8, line 26-30)

The national eye care plan⁴ emphasized activities to reduce blindness focusing cataract surgery that is low-cost, organizing outreach camps for screening, awareness creation, and manpower training. The plan facilitated establishment of treatment centers at district level, and eyesight testing through partnership of government and non-governmental organizations.

7. P6 – 7: sampling: what the census used? Are all people from the country over 40 years old in the sample framework?

>>The study used sampling frame of national Census 2011 for locating the primary sampling units and households. However, the eligible members (aged 40 and older) for the survey were listed during the survey. One out of them were selected randomly. (page 5, line 1-5)

8. How many clusters/size of clusters were selected for the 7,200 people.

>> 72 clusters (primary sampling units) were used. 100 people were selected from each cluster. (page 5, line 4)

9. P9, line 36-40 and P13, table 1, p-value should be given for each of the characteristics to tell the significance.

>> The characteristics were given just to provide the background of the participants. There is no hypothesis behind this. Therefore, we humbly submit that adding p-value to any variable is not necessary. Such approach will invite a problem of multiple testing. As you know, with a set alpha of 0.05, 1 in 20 such tests may appear statistically significant due to chance alone*. Better we avoid such p values here.

*1. Ramganathan P, Pramesh Cs, Buyes M. Common pitfall in statistical analysis: The perils of multiple testing. Perspectives in Clinical Research 2016;7:106-7. doi: 10.4103/2229-3485.179436;

2. Armstrong RA. When to use Bonferroni correctio. Ophthalmic Physiol Opt 2014;34:502-508. Doi: 10.1111.opo.12131

10. P9, line 47-53, these percentage should have confidence interval

>> 95% confidence intervals added

11. P10, discussion: as the age specific prevalence of BL/LV were not presented, so the comparison of prevalence due to ageing is not clear.

>>Age-specific prevalence was presented in the Results section (page 8, lines 9-13, and in Table 2) but for two major groups. We have added **Figure 2** to show age-specific prevalence to comment that blindness increases with age as reported by others (24, 27). (page 8, line 2-4).

In addition, we have added the following sentences for low vision also. (page 9, line 9-13)

Apparently, we observed a higher prevalence of low vision (12.1%) compared to studies in India (9.3%)²⁴, Pakistan (3.3)²¹, Iran (4.0%)²⁷, but it was somewhat similar to that reported from South American countries (5.9 -18.7%).^x These differences should be cautiously interpreted because variation in age composition of the respondents, and some other factors, is an important determinant of low vision.

12. P10, line 27-28, your prevalence was 12.1%, how could this be the same as the countries listed? With less than 1%?

>>Our claim of similarity is based on blindness prevalence, which is 1.0% in our study. (page 8, line 33-34) The 12.1% rate is for low vision. Kindly see clarifications above.

13. P10, line 35 -41, Cataract is the major cause of blindness, the reasons of low prevalence in other countries that the authors given could not convince readers.

>> Please see the response above: the prevalence in other countries is almost same as ours.

14. In general, the discussion is not robust.

>>Because cataract is the major cause of blindness and low vision, we have added a paragraph on **cataract**: (page 9, line 30-35 continued to page 10, line 1-2)

Cataracts attribution to blindness in our sample (79.6%) is a little higher than that reported in an India population (62.1%). Therefore, addressing cataract will be bring most benefit to prevent blindness. In addition to promotion of healthy ageing, a few other factors such as ultraviolet ray exposures, diabetes, hypertension, use of certain drugs, and smoking can be considered. Accessibility to socioeconomically deprived people especially in remote areas should be enhanced. Blindness prevention programe's success will largely depend on the

health system's capacity building to deliver low-cost surgeries. Supplementation from outreach screening will be valuable.

VERSION 2 – REVIEW

REVIEWER	You, Qisheng Oregon Health & Science University
REVIEW RETURNED	11-Dec-2021

GENERAL COMMENTS	The authors have fully addressed the reviewers' concerns and comments.
--

REVIEWER	Xiao, Baixiang Zhongshan Ophthalmic Centre, Sun Yat-sen University, State Key Laboratory of Ophthalmology
REVIEW RETURNED	24-Nov-2021

GENERAL COMMENTS	All the questions were answered and the article is modified appropriately.
--